# OpenReview forum: "Inducing Faithfulness in Structured Reasoning via Counterfactual Sensitivity"
_ICLR.cc/2026/Conference — ICLR 2026 Conference Withdrawn Submission_

### Official Review · Reviewer_RQoa · 2025-10-26

**Soundness:** 1
**Presentation:** 1
**Contribution:** 1
**Rating:** 2
**Confidence:** 4

**Summary:**

This paper introduces a new loss term, CSR, which measures the difference between a model's probability of predicting the same answer under altered traces where the alterations are purposefully meant to cause significantly different answers (such as changing "20 - 8 = _" to "20 + 8 = _").  Focusing on "structured reasoning" domains such as math and logic, the authors create methods for editing traces by changing operators commonly used in the answers for these problems. They then create "verifiers" that determine if the alterations would invalidate the answer given by the model (i.e. should make the answer predict an incorrect answer).  The CSR loss is then meant to encourage the model to predict wrong answers with wrong traces. Evaluations span multiple models and datasets, showing a degradation in accuracy but an increase in "COS," which measures how often models change their answers given altered traces.

The results, however, are difficult to interpret. Many key concepts are left unexplained (including what these "verifiers" are).  There are missing results and tables, and the tables that are present often average across many models and/or tasks making it difficult to understand the effects of CSR.  Section 5, the results, is riddled with undefined metrics, baselines, and experimental design making the findings difficult to trust and understand. It is possible that the majority of this confusion could be resolved by looking through the appendix; however, I doubt this as there are experiments referenced in the main body of the paper that point to the appendix for details but are not actually there. Without this, it's hard to judge if I want a weaker model just so my model answers wrongly to altered reasoning (This isn't really the "faithfulness" we want, it's fixing a symptom of unfaithful models.)

**Strengths:**

I like the idea of providing minor alterations to key parts of the reasoning trace that should yield different answers, and the loss seems relatively sound for experiments.

It appears as though there was a lot of work done! Tons of models and benchmarks. Unfortunatley it's just hard to read / trust them in the current paper (see weaknesses).

**Weaknesses:**

- Section 5 is almost impossible to understand. Every metric you measure, with the exception of accuracy, should be introduced. All models and tasks should be explicitly stated "gpt-3.5-class models" is too vague.  There are missing definitions for validators, "SUFF/COMP", ECE, Operator precision, etc.
- There is very little to no details about how or what you are training. Are you training on the tasks? Is every model/task result an experiment where you train that model on that task? How did you train the editor model? (Obviously, a lot of work was done, so saying everything in the main body is not possible. That being said -- an effort should be made to make this *as clear as possible* for someone reading only the main body of the paper.)
- The writing feels very much like an LLM. Especially because there are pointers to things that do not exist (appendix E does not have human eval nor full per-model evaluations for example)

I really think the writing in section 5 hurt the paper and overall trust of the work reported. It needs to be rewritten extensively without forward references, clearer definitions, and without mentioning experiments / results in the appendix without introducing them (do not discuss results in the main body of the paper without introducing stuff, pointing to the appendix is confusing and hurts trust).

If these concerns were fixed and section 5/results were made clearer, I think there's a more foundational weakness which is the loss is leading to models with lower accuracy but higher COS. However, the CSR loss is really just the COS metric (so you are training models on the metric you want to your model to have).  This is fine if we all agree that COS is the end all metric for good faithful models (or at least that it leads to more reliable reasoning). Which I believe is why section 4 is trying to provide theorems for it, but they are not convincing. 1 because it's partial and hard to understand without reading the appendix and 2 because it's not paired with human eval which I think would be necessary for this kind of claim.

**Questions:**

See weaknesses, I have many questions that I expect a more careful rewrite of section 5 would solve.

---

### Official Review · Reviewer_Joi7 · 2025-10-30

**Soundness:** 1
**Presentation:** 1
**Contribution:** 2
**Rating:** 0
**Confidence:** 4

**Summary:**

This paper tackles the problem of faithfulness in chain-of-thought reasoning: prior work has shown that language models can be insensitive to the reasoning traces they generate, showing that they do not necessarily rely on the trace to derive their final answer. This paper proposes a theoretical framework for thinking about faithfulness and introduces Counterfactual Sensitivity Regularization (CSR), a training method that encourages models to be sensitive to their traces. Results on a variety of benchmarks using Llama 2 8B show improved output sensitivity with minimal impact on final answer accuracy.

**Strengths:**

The paper tackles an interesting problem and proposes to be theoretically grounded. The idea of measuring sensitivity to outputs and using it as a regularization during training is quite interesting and potentially promising.

**Weaknesses:**

The paper is unfortunately extremely vague and inconsistent in several aspects, both in its theoretical part and in the experiments.

* First, the paper emphasizes that "The core of CSR’s effectiveness lies in the quality of its counterfactual traces.". This is accomplished via a "learned editor model", which is completely deferred to Appendix C. Then, Appendix C gives no detail on this editor model. The model is a small transformer - trained from scratch purely from RL? It seems like it outputs "sequence of edit operations". If it starts from scratch, it will likely start with random edits. If not from scratch, then the paper should describe how it is initialized, then. The current description leaves me very suspicious.
* The paper seems to suggest a simple "compute loss, add regularization, take gradient step" training loop (e.g., see Algorithm 1). But if training involves generating reasoning traces, then that is not how it works. You'd need some RL method, like STaR (filter + SFT), GRPO, etc.
* The abstract mentions "8.7% training overhead". This number seems low given the procedure involves editing a reasoning trace and doing other forward passes. Even then, 8.7% does not show up in the results anywhere (Table 4 has an "overall" of 9.2%).
* The "THEORETICAL FOUNDATIONS" section is extremely vague. None of the "Definitions" are presented as mathematical definitions. The text says that the full definitions are in the appendix, but they are not. The appendix only has "proofs" of the theorems. But without definitions, there can be no proofs about them.
* The results are also poorly explained. None of the baselines are defined. GPU hours seem highly inflated (you don't need that much compute on GSM8k). Also, what GPU is this, anyway? GPU hours without the GPU model is a completely meaningless metric. Accuracy results are bolded for the CSR models, which have the worst accuracies. The absolute numbers also don't make much sense -- Llama 2 did not perform that well on GSM8k (the Code LLama paper reports only 24% for LLama 2 13B on GSM8k, while here somehow all methods get 80%+).

In summary, the paper unfortunately does not have nearly enough detail on either the method or the results, and this makes me highly suspicious of whether the results are real.

**Questions:**

Please clarify as much as possible the points in the weakness.

---

### Official Review · Reviewer_9nXZ · 2025-10-31

**Soundness:** 3
**Presentation:** 3
**Contribution:** 3
**Rating:** 4
**Confidence:** 4

**Summary:**

This paper studies the problems of generating fake CoT in current reasoning models. This is due to outcome-only training that rewards only the final answer. To solve the problem, the paper introduces Counterfactual Sensitivity Regularization (CSR), a training-time objective to induce faithful CoT in LLMs. Specifically, for each input X, the authors first use a model to generate a reasoning trace T and then generate the answer Y. After that, a learned editor minimally perturbs T to a counterfactual CoT T'. Then, CSR maximizes the divergence between the original and counterfactual answer.

The authors also provide the theoretical foundations of their method. Then, they tested their loss on seven different datasets. The results show that their method can improve the Counterfactual Outcome Sensitivity by a large margin.

**Strengths:**

1. The paper studies a very important problem of generating fake CoT for current LLMs.
2. The authors verified their methods on 7 different datasets.
3. The authors also provide the theoretical foundations of their method.

**Weaknesses:**

1. The models saw a lot of wrong CoT(s) (the T') in this method, which are generated by the learned editor model. Such wrong CoT(s) may affect the performance of the tuned models. Also, in Table 1, I find the Acc scores of the proposed method are lower than baselines, which further supports my concerns.

2. The method needs a learned editor model, which is hard to scale. The authors need to train a editor model for every single task.

3. In Section 5.6 Efficiency, is the time of training the editor model included in the time cost of the CSR method? This may introduce unfair comparison.

**Questions:**

See the weakness part.

---

### Official Review · Reviewer_SRZQ · 2025-10-31

**Soundness:** 1
**Presentation:** 2
**Contribution:** 1
**Rating:** 2
**Confidence:** 4

**Summary:**

The paper aims to enforce faithfulness in model reasoning by penalizing models when corrupted reasoning traces still produce correct answers. The paper introduces CSR a method to force LLMs to have a dependence between their reasoning trace and the outcome: if the process of arriving at an outcome is changed at an intermediate point then the outcome must change too. To do this, the authors train a perturbation model that makes instrumental edit and then train models with this editor so that the reasoning is causally linked to the final answer.

**Strengths:**

- I like the approach of the learned editor to introduce perturbations in the reasoning.
- The authors do a good job of evaluating a wide range of domains/ tasks

**Weaknesses:**

- There is no metric provided that dirrectly meaasure the premise of the problem of the paper: “a model may generate a correct answer while relying on a flawed or irrelevant reasoning
trace.” meaning the model **naturally produces unfaithful reasoning**. However, COS measures something entirely different: sensitivity to **artificially introduced** noise via operator-level edits.
    - **COS**: Does the model change its answer when we manually corrupt a valid reasoning trace?
    - **What the problem requires**: Does the model naturally generate flawed reasoning while producing correct answers?
    - Without measuring the actual problem, we cannot know if CSR solves it. The improvements in COS (sensitivity to synthetic corruption) don't demonstrate reductions in naturally occurring unfaithful reasoning. The paper needs to show that CSR-trained models generate fewer instances of "correct answer + flawed reasoning" in normal operation, not just that they respond differently to artificial perturbations.
- The paper does not address how CSR handles reasoning traces where the model abandons a path and backtracks. If a model explores multiple solution strategies and abandons incorrect ones, counterfactual interventions in the abandoned portions should NOT change the final answer. CSR would penalize this as “unfaithful.” Or is  this addressed by a learned edit model.
- The primary experiments use **Llama-2-13B**, which is a pretty outdated model.
- PRM, verifier guided training seem outdated. Other methods like GRPO would make the experiments more relevant. Especially since most experiments use gpu-hours as a central metric.
- The overall setup of an external model perturbing the reasoning trajectory is pretty unrealistic. Models usually perform terribly when low probability tokens are placed in their output (off-policy).
- CSR Usually leads to a slightly lower overall accuracy.  The paper doesn't adequately justify why we should accept accuracy losses for improved COS.

**Minor:**

- 253: "four flagship benchmarks" is awkward phrasing (why "flagship"?)
- Some notation is introduced but not consistently used (Definition 1)
- The learned editor training is not described in detail.

**Questions:**

See the weaknesses.

---

### Note · Authors · 2025-11-12

**Comment:**

Thanks, reviewers, for the detailed feedback. There were errors that we overlooked, and a major revision is needed. Hence ,we are withdrawing the paper.

**Withdrawal Confirmation:**

I have read and agree with the venue's withdrawal policy on behalf of myself and my co-authors.